# Extension of the Voronoi Diagram Algorithm to Orthotropic Space for Material Structural Design

**DOI:** 10.3390/biomimetics9030185

**Published:** 2024-03-19

**Authors:** Pavel Bolshakov, Nikita Kharin, Alexander Agathonov, Evgeniy Kalinin, Oskar Sachenkov

**Affiliations:** 1Institute of Mathematics and Mechanics, Kazan Federal University, 420008 Kazan, Russia; pavvbolshakov@kpfu.ru (P.B.);; 2Department Machines Science and Engineering Graphics, Kazan National Research Technical University named after A.N. Tupolev, 420111 Kazan, Russia

**Keywords:** structural design, porous constructions, structural material, orthotropic material, Voronoi diagram

## Abstract

Nowadays, the interaction of additive technologies and methods for designing or optimizing porous structures has yielded good results. Construction with complex microarchitectures can be created using this approach. Varying the microarchitecture leads to changes in weight and mechanical properties. However, there are problems with geometry reconstruction when dealing with complex microarchitecture. One approach is to use Voronoi cells for geometry reconstruction. In this article, an extension of the Voronoi diagram algorithm to orthotropic space for material structural design is presented. The inputs for the method include porosity, ellipticity, and ellipticity direction fields. As an example, a beam with fixed end faces and center kinematic loading was used. To estimate robust results for different numbers of clusters, 50, 75, and 100 clusters are presented. The porosity for smoothed structures ranged from 21.5% up to 22.8%. The stress–strain state was determined for the resulting structures. The stiffness for the initial and smoothed structures was the same. However, in the case of 75 and 100 clusters, local stress factors appeared in the smoothed structure. The maximum von Mises stress decreased by 20% for all smoothed structures in the area of kinematic loading and increased by 20% for all smoothed structures in the area of end faces.

## 1. Introduction

Developing methods for modeling and designing custom endoprostheses is one of the important tasks of personalized medicine [1,2,3]. Additive manufacturing offers many advantages in the production of high-quality, patient-specific porous implants [4,5]. The application of complex unstructured geometry makes it possible to obtain new product properties that provide improved treatment and improve the quality of patient rehabilitation [6]. Such structures may not only have unique mechanical properties but also new biological properties [7,8,9]. Thus, modern biomedical engineering is aimed at designing and producing patient-specific devices with distinctive regenerative properties. This regenerative effect can be achieved by adding a pharmacological agent to the chemical composition of the material [10,11,12]. Despite the development of the use of biocompatible materials and complex structured products, issues related to design methods and technological features, such as the quality of the resulting surfaces, require further study [13,14,15].

However, amazing examples of optimal designs are observed in nature. A good example is bone. The internal structure of bone adapts to external loads while considering mass reduction. However, this structure is quite complex. From the point of view of solid mechanics, it can usually be described as an orthotropic, asymmetric, and heterogeneous medium. Thus, the goal of biomimetic design is to recover such a structure for artificial products. Nowadays, the possibility of manufacturing such complex samples rests on additive manufacturing, and there is a lot of research in this area. Some methods are based on copying existing structures, whereas others attempt to copy the principles of structure formation. Therefore, research into the design of such products is one of the cutting-edge areas of modern engineering [16]. First of all, we can distinguish topological optimization methods [17,18]. This approach modifies the boundary of the design volume according to the target function. Separately, it is possible to allocate the approach in which the dimensionality of the design model is reduced, and the corresponding geometrical parameters are subjected to modification [19,20]. An alternative approach involves changing the geometry (e.g., pore distribution) within the computational domain. Pore geometry is often described using representative volumes. In this case, each representative volume contains a parameterization of its microarchitecture. In this approach, we can separately address the problem of geometry reconstruction from the obtained distribution of representative volume parameters. The obvious one is direct restoration, but this approach is labor-intensive, and the results are often not technologically advanced [21]. An alternative is the use of Voronoi cells for geometry restoration [22,23]. From a biomimetic point of view, Voroni structures hold promise for life. Similar structures are observed in foliage and insect tissues. Researchers have reported positive results regarding Voronoi scaffolds in terms of mechanical [24,25] and biomechanical parameters [26]. With this approach, we can understand the computational domain as a domain on which some scalar field is defined. For this domain, it is possible to construct weighted Voronoi cells (taking into account the value of the scalar field as a weight) [27,28]. A logical extension is the development of optimization methods using Voronoi cells. In [29], a connection between microstructure and macroscopic scale density was found. Subsequently, the optimization and reconstruction problems were solved for the full-scale graded lattice structure. In [30], the biomimetic capabilities of the Voronoi-based cancellous bone microstructure were investigated. A new parametric method for the design of Voronoi-based lattice porous structures is presented in [31]. The method was based on the relationship between parametric microstructure and macroscopic scale density. This approach assumes a specific microstructure architecture. Additionally, it should be mentioned that the resulting microarchitecture is isotropic in macroscale terms [29,30] or is gradient-based [31].

The aim of this study was to generalize the Voronoi diagram method to the case of locally orthotropic space for structure reconstruction problems with microarchitecture.

## 2. Materials and Methods

### 2.1. Brief Description of the Proposed Design Method

The concept of previously developed structural design revolves around the idea of a basic cell. According to this idea, the geometry’s volume is considered a subset of basic cells, which can be represented as follows:(1)$$\overset{N}{\underset{i=1}{⋃}}V_{i}^{BC}=V$$
where *V^BC^* represents basic cells.

Basic cells should not intersect each other. This can be represented as follows:(2)$$\overset{N}{\underset{i=1}{⋂}}V_{i}^{BC}=\emptyset$$

Each basic cell possesses some mechanical properties. Previously, we considered them as anisotropic media [32,33]. Assuming that the origin of anisotropy is provided by the microarchitecture, the parameter vector ***p*** was introduced. So, the effective stiffness tensor can be presented as:(3)$$\tilde{\tilde{C}}=\tilde{\tilde{C}}(\vec{p}\left(\vec{x}\right))$$

Wherein the parameter vector and, as a consequence, the stiffness tensor are constant in the domain of each basic cell:(4)$$\tilde{\tilde{C}}\left(\vec{p}\left(\vec{x}\right)\right)={\tilde{\tilde{C}}}_{i},\;\forall \vec{x}\in V_{i}^{BC}$$

The task of structural design is to find the distribution of the parameter vector for a certain type of microarchitecture. The widespread assumption is the existence of material symmetry in basic cells. Orthotopic symmetry is a common type of symmetry. So, let us consider a plane problem and orthotopic basic cells. In this case, the parameter vector can be defined as a set of ellipticity *β*, porosity *η*, and main direction of an ellipticity ***e***:(5)$$\vec{p}(\vec{x})=\left(\beta (\vec{x}),\eta (\vec{x}),{\vec{e}}^{T}(\vec{x})\right)$$

This parameter vector is defined at each point of a study volume. The polynomial connection between stiffness tensor components and ellipticity and porosity was found previously, as follows [32]:(6)$$C_{kl}\left(\lambda ,\beta \right)=\sum_{i=0}^{3}\sum_{j=0}^{3}c_{ij}\lambda^{i}\beta^{j}$$

*C*_*kl*_ in Equation (6) corresponds to Young’s modulus, the shear modulus, and Poisson’s ratio in orthotropic directions. Non-zero polynomial coefficients (*c*_*ij*_ in Equation (6)) are shown in Table 1.

Using Equation (6), the structural design problem can be solved. As a result, the distribution of the parameter vector in the geometry’s volume can be found. The next step is to restore the resulting geometry, which should provide the vector field. The evident solution is to draw each basic cell directly. In practice, a finite element mesh is associated with a distribution of basic cells. This means that each finite element is treated as a basic cell. This is generally not true. Additionally, the problem of smoothing the microarchitecture on the borders of the basic cells occurs. From this point of view, the shape of each basic cell is unknown. One of the solutions is to use the distribution of the parameter vector to solve this problem. This study used the clustering method for porosity distribution. Next, the pore shape should be restored in each cluster according to the distribution of ellipticity values and directions. For these purposes, the Voronoi diagram for orthotropic space was generalized.

### 2.2. Mathematical Formulation of the Smoothing Problem

Let us consider the results of the distribution of the parameter vector as a result of the design task. The next problem is to restore the corresponding geometry. Then, the geometry’s volume can be presented as a set of subvolumes *h*_*i*_, as follows:(7)$$\overset{N}{\underset{i=1}{⋃}}h_{i}=V$$

With the obvious condition of
(8)$$\overset{N}{\underset{i=1}{⋂}}h_{i}=\emptyset$$

Considering *h*_*i*_ as a cluster set by porosity distribution:(9)$$\langle \eta \left(\vec{x}\right)\rangle =M_{i},\;\vec{x}\in h_{i}$$
(10)$$\left|M_{i}-\eta \left(\vec{x}\right)\right|\leq \varepsilon ,\vec{x}\in h_{i}$$
(11)$$\left|M_{i}-\eta \left(\vec{y}\right)\right|>\varepsilon ,\vec{y}\notin h_{i}$$

This means that the markup function can be defined as follows:(12)ϕ:V→Z

Considering Voronoi cell as
(13)$$V\left(T\right)=\left\{p:\forall \upsilon \in T\;\omega \in S/T,\;d(p,\upsilon )<d(p,\omega )\right\}$$

The order-*k* Voronoi diagram can be defined as follows:(14)$${Vor}_{k}\left(S\right)=⋃V(T),T\subset S,\left|T\right|=k$$

The difference from a classic approach is the orthotropic property of the metric space. This leads to modification of the distance function. For each point, principal directions can be understood as a local coordinate system, and ellipticity can be understood as the scale factor for an axis. So, the Euclidean distance with the corresponding scale factor can be used.

An algorithm for obtaining a generalized Voronoi diagram on orthotropic space has been developed and is provided below (smoothing structure algorithm). The inputs for the algorithm include mesh and parameter vector (*β*, *η*, and ***e***) distribution, the number of clusters *K*, and Voronoi mesh parameters (*N*_dim_ and *M*_dim_). The outputs were markup functions defined as binary maps and clusters data.

The flood fill algorithm for segmented data (see Algorithm 1, smoothing structure algorithm) was modified for this purpose. To take into account the local ellipticity and its direction, a Bresenham circle with a radius of 1 was used. To modify the flood fill algorithm, we introduced the weight at the point. This is an intermediate parameter before filling with the target color (filling by color if weight was higher than the critical value, denoted as s in the ColorStep algorithm). To improve accuracy, 8 directions were used in the modified algorithm (see Algorithm 2, ColorStep algorithm).
**Algorithm 1.** Smoothing structure algorithm**Inputs:** mesh, *β*, *η*, ***e***, *K*, *N*_dim_, *M*_dim_**Outputs:** Binary map: clMap; clusters: Kmap*newmesh* ← generate regular mesh (*N*_dim_ × *M*_dim_) in the mesh domain*newmesh* ← interpolated data of the parameter vector*newmesh* ← clusterization of *newmesh* by η on K clusters*color* ← unique (*Kmap*)clMap ← 0**For each** *cluster* **in** *Kmap*        clMap (*cluster.center*) = s        stPoint = *cluster.center*        clMap ← ColorStep(clMap, *newmesh*, *cluster*, stPoint, *color*; eps, s)**end for**clMap (clMap ≥ s) ← 1clMap (clMap < s) ← 0

**Algorithm 2.** ColorStep algorithm**Inputs:** mapColor, mapData, clusterN, point, color; eps, s**Outputs:** mapColori ← point.ij ← point.j**e** ← mapData.mainDirection(i, j)**e** → direction: {[0, 1]; [1, 0]; [1, 1]; [−1, 1]}tangent ← map(direction): {[1, 0]; [0, 1]; [−1, 1]; [1, 1]}diag ← map(direction): {[1, 1] and [−1, 1]; [1, 0] and [0, 1]}ptemp ← size (mapColor == s **and** mapData.cluster = color(clusterN))ptemp ← ptemp/size(mapColor **and** mapData.cluster = color(clusterN))β ← mapData.ellipticity(i, j)p ← mapData. cluster(clusterN).meanPorosity**If** |p(*cluster*) − ptemp|/ptemp > eps        break
**end if**
**If** mapColor [(i,j) ± direction] ≠ s **and** mapData.cluster = color(clusterN) **then**mapColor [(i,j) ± direction] += 1
**else**
        colorStep(mapColor, (i,j) ± direction, β, **e**, s)
**end if**
**If** mapColor [(i,j) ± tangent] ≠ s **and** mapData.cluster = color(clusterN) **then**mapColor [(i,j) ± tangent] += β**else** 
*break*
        colorStep(mapColor, (i,j) ± tangent, β, **e**, s)
**end if**
**If** mapColor [(i,j) ± diag] ≠ s **and** mapData.cluster = color(clusterN) **then**mapColor [(i,j) ± diag] += sqrt(2β^2^/(β^2^ + 1))
**else**
        colorStep(mapColor, (i,j) ± diag, β, **e**, s)
**end if**


The algorithm is illustrated in Figure 1. An example of a 5 × 5 mesh was considered, with a constant ellipticity value of 0.5 and a critical weight value equal to 3. In Figure 1a, the first step of the algorithm is illustrated. The initial cell (a cluster center) was filled with cyan color, and the ellipticity vector is shown by an arrow. Then, according to the algorithm, the cells were filled in a Bresenham circle with a radius of 1. Thus, in Figure 1b, the cells in the ellipticity direction were colored in red (with a weight equal to 1), the cells in the orthogonal direction were colored in blue (with a weight equal to 0.5), and the cells in other directions were colored in light green (with a weight equal to 0.63). After three iterations, the weight of the cells accumulated, and critical values appeared in the red cells (see Figure 1c). This cell was colored according to the algorithm’s terms. Therefore, cyan and two red cells were added to the list and, for each cell in the list, the same procedure in a Bresenham circle with a radius of 1 was repeated. In Figure 1d, the next iteration is illustrated. The ellipticity direction is shown by an arrow (see Figure 1c). In Figure 1d, the weights are shown after another iteration step. In the ellipticity direction, the weights were incremented by a value of 1. In the orthogonal direction, the weights were incremented by a value of 0.5. In the other directions, the weights were incremented by a value of 0.6. The total weights are presented in Figure 1d. Such iterations were repeated until the count of colored cells was less than the limit. The limit value was calculated using porosity and cluster area (volume) values. The colored cells were interpreted with different meanings. In this study, the colored cells represented pores.

### 2.3. Test Task

To test the proposed algorithm, test tasks were performed. For this purpose, one cluster was investigated. Different values of porosity, ellipticity, and the direction of ellipticity were used to reconstruct the Voronoi cell. The porosity values varied from 0.1 to 0.4 in steps of 0.05. The ellipticity value varied from 0.1 to 1 in steps of 0.05. The direction value varied from 0° to 90° in steps of 5°. The smoothing structure algorithm was implemented in Matlab R2019a software.

### 2.4. Model Task

A rectangular plate of 140 mm × 28 mm × 14 mm (see Figure 2) was used for implementation of the algorithm. The mechanical behavior of the rectangular plate (region *V* in **R**^3^) with the boundary ∂*V*, within the linear theory of elasticity, can be described by the following system of equations:(15)$$\vec{\nabla }\cdot \tilde{\sigma }=0,\;\forall \vec{x}\in V^{0}\subset R^{3}$$
(16)$$\tilde{\varepsilon }=\frac{1}{2}\left(\vec{\nabla }\vec{u}+{\left(\vec{\nabla }\vec{u}\right)}^{T}\right),\;\forall \vec{x}\in V^{0}\subset R^{3}$$
(17)$$\tilde{\sigma }=\tilde{\tilde{C}}:\tilde{\varepsilon },\;\forall \vec{x}\in V^{0}\subset R^{3}$$
(18)$$\vec{u}=0,\;\forall \vec{x}\in S_{F}$$
(19)$$\vec{u}={\vec{u}}_{0},\;\forall \vec{x}\in S_{A}$$
(20)$$\tilde{\sigma }\cdot \vec{n}=0,\;\forall \vec{x}\in \partial V\backslash \left(S_{A}\cup S_{F}\right)$$
where *V*° = *V* ∪ ∂*V*; ***u*** is the displacement vector; σ is the stress tensor; ε is the elastic strain tensor; and *C* is the stiffness tensor. *S*_*F*_ is the surface with no displacement, and *S*_*A*_ is the surface on which kinematic boundary conditions were specified. ***n*** is a normal to the corresponding surface.

The problem (15)–(20) was solved using the finite element method. Eight-node hexahedral finite elements were used for the calculations. A kinematic loading of 1 mm was used in the numerical simulation. The length of the kinematic loading region was 20 mm. So, the region and boundary condition in (15)–(20) can be specified as follows:(21)$$V^{0}:x\in \left[-70,70\right],y\in \left[-14,14\right],z\in \left[-7,7\right]$$
(22)$$S_{A}:\;x\in \left[-10,10\right],y=14,z\in \left[-7,7\right]$$
(23)$$S_{F}:\;x\in \left\{-70,70\right\},y\in \left[-14,14\right],z\in \left[-7,7\right]$$
(24)$${\vec{u}}_{0}=\left(0,-1,0\right)$$

According to the method developed in [32], the region was divided by two parts: the region with and without structural modification. Structural modification leads to changes in the stiffness tensor values. So, Equations (15)–(24) should be extended as follows:(25)$$V=V_{C}\cup V_{D}$$
(26)$$V_{C}\cap V_{D}=\emptyset$$
(27)$$\vec{p}(\vec{x})=const\;\Rightarrow \;\tilde{\tilde{C}}(\vec{x})=const,\;\forall \vec{x}\in V_{C}$$
(28)$$\vec{p}(\vec{x})\neq const\Rightarrow \tilde{\tilde{C}}(\vec{x})\neq const,\;\forall \vec{x}\in V_{D}$$
where *V*_*C*_ is a region without structural modification and *V*_*D*_ is a region with structural modification.

The loading scheme is presented in Figure 2, where the region *V*_*C*_ (with no structural modification) is marked in green. The end faces of the beam were fixed (18) and (23). Kinematic loading was applied to the middle of the plate (19) and (22). The parameter vector distribution was found using the method described in [32]. The stiffness tensor component values (28) were calculated using the structure parameters in Equation (6) and the polynomial coefficient values in Table 2.

As a result of the method developed in [32], the distribution of parameter vector ***p*** in region *V* was found. The parameter vector consists of the ellipticity value, the porosity value, and the main direction of ellipticity. Then, the computational domain was remeshed using a Cartesian fine grid (with each edge sized to about 6/100 of the computational grid). The parameter vector field was smoothed on the Cartesian grid. The parameters for smoothing were as follows: eps = 0.05 and s = 3. The region *V* was clustered by porosity distribution. Three variants of clusterization were used for smoothing: 50, 75, and 100 clusters. As a result, the segmentation data was obtained. The STL file was restored using the segmentation data. This STL file can be used for both direct simulation of the stress–strain state and for manufacturing. The reconstructed smoothed geometry was simulated in the same problem formulation (15)–(24). Direct simulation of the stress–strain state was performed using FEM. Ten-node tetrahedral finite elements with quadratic approximation were used for the direct simulation. The stress and strain field distributions for the original and smoothed beam were compared. The smoothing structure algorithm was implemented using Matlab software. The stress–strain problem was solved using Ansys v. 14 software.

## 3. Results and Discussion

### 3.1. Test Task Results

The results for the region with 32 pixels per side are shown in Figure 3. The samples in the columns in Figure 3 have constant ellipticity values of 1, 0.7, 0.5, and 0.2, respectively. The samples in the rows in Figure 3 have constant porosity values of 0.2, 0.3, and 0.4, respectively. The results for different ellipticity directions are shown in Figure 3a, Figure 3b, and Figure 3c.

Some form deviations were noticed for the spherical pores (ellipticity equal to one) with porosity values below 0.2. In this case, some hairy pixels were observed (see Figure 3, with an ellipticity of one and a porosity 0.2). The number of hairy pixels decreased when the ellipticity was not equal to one (see Figure 3, with a porosity of 0.2).

The form became more accurate as the porosity increased. A representative picture is shown in Figure 3, with a porosity of 0.3. Hairy pixels occured only when the ellipticity was 0.2 and the ellipticity direction was 45°.

The last porosity value was 0.4. For low ellipticity, in this case, form limitations appeared due to cluster borders (see Figure 3, with an ellipticity of 0.2 and a porosity of 0.4).

From Figure 3, it is clear that the received forms for cases with an ellipticity direction of 0° and 90° were equal within rotation. When the ellipticity direction was 45°, there were some deviations. This can be explained by pixel filled density. In the test task, grids with 32 by 32 pixels were used. Increasing this value, on one hand, improves form quality, but on the other, leads to increasing required memory. This is not important for test tasks but can be crucial for real problems.

### 3.2. Lightening of the Original Construction

The initial geometry was divided by two regions: with and without structural modification (represented by the blue and green regions in Figure 2, respectively). An eight-node hexahedral finite element was used for the simulation. The finite element size for the region with structural modifications was 4 × 4 × 4 mm. The finite element size for the region without structural modifications was 2 × 4 × 4 mm. Then, the structural design was performed.

The stress–strain state for the initial (solid) and the structurally designed plate were compared. The Cartesian cell size for smoothing was equal to 0.25 mm. The cell size was defined by the factor of production. The sample porosity was equal to 21%. The critical accuracy for the porosity values in clusters via smoothing was equal to 5%. So, smooth geometry was built for three clusterization variants. Clusterization was performed using porosity distribution. A total of 50, 75, and 100 clusters were used. The sample porosity for 50, 75, and 100 clusters was equal to 22.8%, 21.8%, and 21.5%, respectively. Compared with the directly restored geometry [32], the smoothed geometry illustrated qualitatively good results. The resulting distribution deviated from the initial distribution. This can be explained by the lack of robustness in the clustering process. The impact of clusterization deviation on the sample porosity was about 2%. The smoothed geometry was meshed for the following simulations. A three-point bending simulation was performed using kinematic loading. Some typical zones were detected. The zones are shown in Figure 4 (the zones are marked using roman numerals I, II, and III), and the porosity values in the zones are provided in Table 2.

First, the displacement filed was analyzed. The displacement on the opposite side of the loading site was about 0.095 mm, compared to 0.1 mm for the solid sample (with a relative deviation of 0.5%). The displacement distribution in the longitudinal direction was almost the same for all the clusterization cases. Of course, some deviations occured, but they were insignificant. The displacement fields are shown in Figure 4.

Then, the von Mises stress field was analyzed. The von Mises stresses on the opposite side of the loading site were about 200 MPa, compared to 240 MPa for the solid sample (with a relative deviation of 20%). Stress raisers obviously appeared in the location of kinematic loading and in the corner of the end wall. Hot spot stress (about 200–240 MPa for 50 clusters) appeared between the pores. This can be explained by local thinning. For 75 and 100 clusters, stress raisers appeared in these parts (box area in Figure 5c,d). The von Mises stress fields are shown in Figure 5.

In Figure 6, the critical zones are shown. For 75 clusters, the stress factor appeared in the arch. This stress factor persisted even for the case of 100 clusters.

## 4. Discussion

The resulting structures (see Figure 4) correlate with the results from another research study [34]. Let us focus on zones. The largest porosity value appeared in zone II (about 15% compared to 2.4–5% in zones I and III). Similar results were obtained in [34], where the largest porosity value appeared in places close to zones I and II. This difference can be explained by the applied boundary condition. Thus, in previous research, the end faces were fixed [32], which led to stress factors in the end face area and additionally resulted in increased porosity in zone I. In zone III (the kinematic loading area) and II, the pore shape and shape direction were similar to the ellipticity distribution from [32] and [34]. Using the porosity distribution in [32,34], the structure was directly reconstructed (see Figure 7a). In this paper, an extension of the Voronoi diagram algorithm to the orthotropic space was implemented for the same distribution. Numerical calculations showed that a stable structure appeared when the number of clusters was more than 50. The clusters of 50, 75, and 100 are shown in different colors in Figure 7b–d. Porosities of less than 1% were not reconstructed. The resulting pores in each cluster are shown in Figure 7b–d. The obtained pore distributions are qualitatively similar for both the direct method and the Voronoi diagram method.

Deviation of the total porosity from the original structure was 8% for the case of 50 clusters and about 2% for the other cases. The resulting designs (Figure 7b–d) are more technologically advanced. In addition, the data obtained can be directly utilized for fabrication by laser stereolithography. On the other hand, local stress factors appeared, which motivates further improvement of the proposed approach. Nevertheless, the developed approach allows for the reconstruction of porosity based on the distribution of the parameter vector.

Similar research [35,36] should be highlighted. In the literature, Voronoi-based structures have been investigated using numerical and full-scale tests. Detected stress factors are a key problem in the research and in general material design. According to numerical results, we cannot be sure about the nature of stress factors (whether they are a numerical or boundary condition problem, or whether they are due to local strength loss). Of course, a large number of calculations with local remeshing can shed light on this issue, but this costs time. So, in [35], a biomimetic approach was used. The initial structure was derived from the wing geometry of the dragonfly *Didymops floridensis*. The algorithm proposed in [35] does not take into account changes in stress distribution during structural formation, and the strength was checked in the full-scale tests after the structure was formed. In [36], an optimized density map was used for initial data for structural composition. However, in [36], the thickness of the Voronoi wall depended on local density. Such an approach indirectly takes into account stress distribution, but only in terms of local isotropy. The transition to Voronoi cells disrupts the isotropy, leading to the emergence of stress factors. Additionally, using only the density map in [36] leads to assuming volumetric stress at each point.

Conversely, the presented approach allows for obtaining a Voronoi cell structure while taking into account both porosity distribution (or the density map) and ellipticity distribution (local stress behavior). Such an approach enables consideration of both principal stresses, leading to the orthotropic properties of the resulting structure. The presented results do not contradict existing results and contribute to a broader understanding of structure design methods. Still, the problem of local strength for the resulting structure exists, and developing a new method that takes into account stress–strain changes during structure formation is a pathway for further development.

## 5. Conclusions

In this article, the extension of the Voronoi diagram algorithm to the orthotropic space for a material’s structural design is presented. Relevant algorithms and numerical results are presented. The input data for the proposed method included porosity and ellipticity fields.

In the numerical results, a beam with fixed end faces and kinematic loading at the center were used. The porosity field was used for clustering. Cases of 50, 75, and 100 clusters were presented. Then, the Voronoi diagram algorithm was extended to orthotropic space. For this purpose, the ellipticity field was used. Local orthotropic directions and ellipticity were used. The corresponding total porosity values were 22.8%, 21.8%, and 21.5%.

The stress–strain state was investigated for the resulting structures. The maximum von Mises stress decreased by 20% for all cases of the smoothed structure in the area of kinematic loading. The maximum von Mises stress increased by 20% for all cases of the smoothed structure in the area of the end faces. In the cases of 75 and 100 clusters, a local stress factor appeared in the smoothed structure. The stiffness for the initial and smoothed structure was the same.

The proposed algorithm can be improved using a Bresenham circle with a radius of more than 1. Some smoothing can be applied to the resulting cell map, which can improve the direct numerical results. This research can be developed for three-dimensional cases. In addition, the problem of clustering should be mentioned. The number of clusters used for smoothing is controversial. More research is needed to find the optimal number or appropriate criteria.

The general pipeline can be formulated to restore geometry. The first step is to formulate the boundary value problem for the stress–strain state. Then, it is necessary to find the parameter vector for this problem (the method given in article [32] or another one can be used). The last step is to apply the proposed Voronoi diagram algorithm to the parameter vector distribution.

## Figures and Tables

**Figure 1 biomimetics-09-00185-f001:**
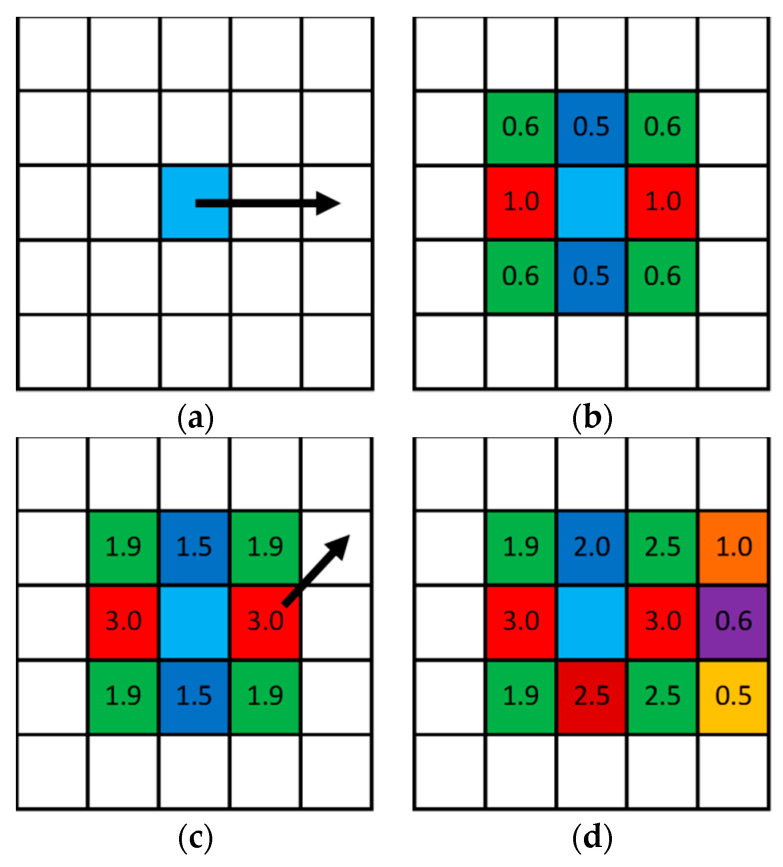
Illustration of the algorithm: (**a**)—initial step, (**b**)—result after 1 iteration, (**c**)—result after 3 iterations, (**d**)—result after 4 iterations.

**Figure 2 biomimetics-09-00185-f002:**
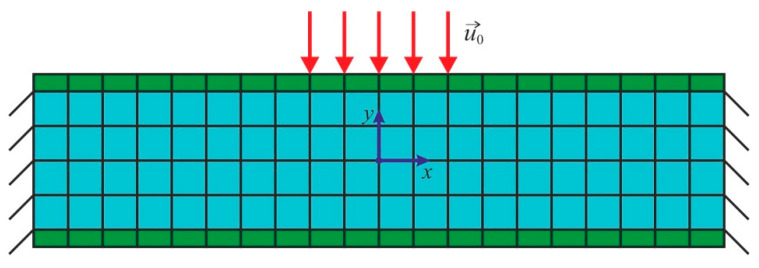
Loading scheme. ***u***_0_ is the applied displacement, the blue region is a region of structural modification (*V*_*D*_), and the green region is a region with no structural modification (*V*_*C*_).

**Figure 3 biomimetics-09-00185-f003:**
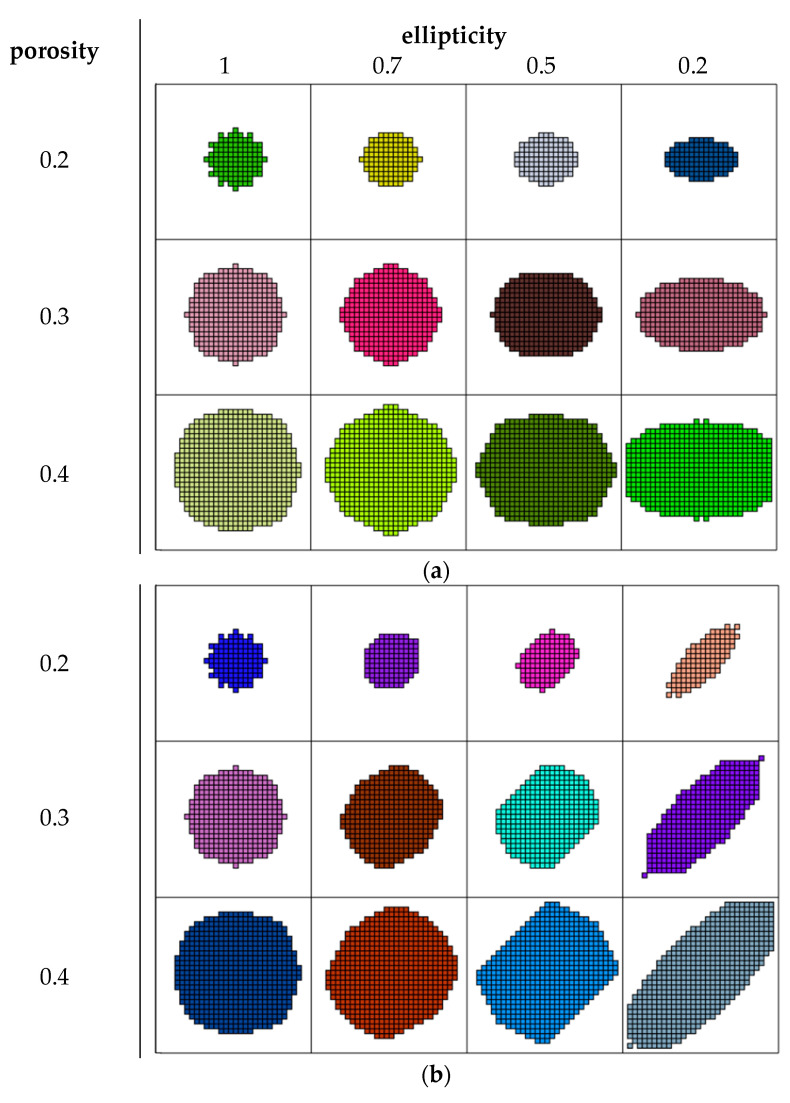
Test tasks results: (**a**)—horizontal ellipticity direction, (**b**)—diagonal ellipticity direction, and (**c**)—vertical ellipticity direction.

**Figure 4 biomimetics-09-00185-f004:**
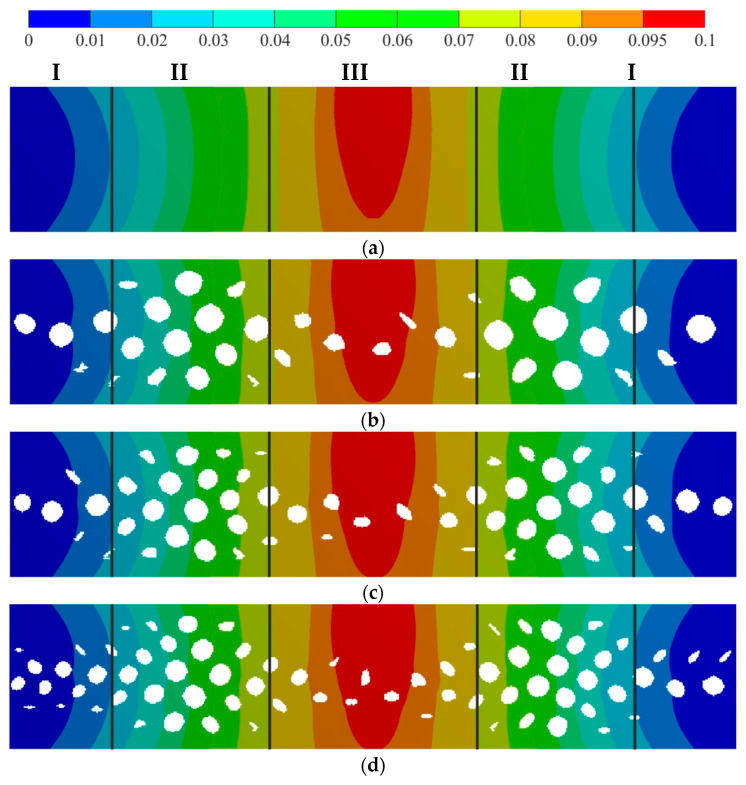
Beam displacement (deflection) field, mm: (**a**)—original structure, (**b**)—smoothed structure for 50 clusters, (**c**)—smoothed structure for 75 clusters, and (**d**)—smoothed structure for 100 clusters.

**Figure 5 biomimetics-09-00185-f005:**
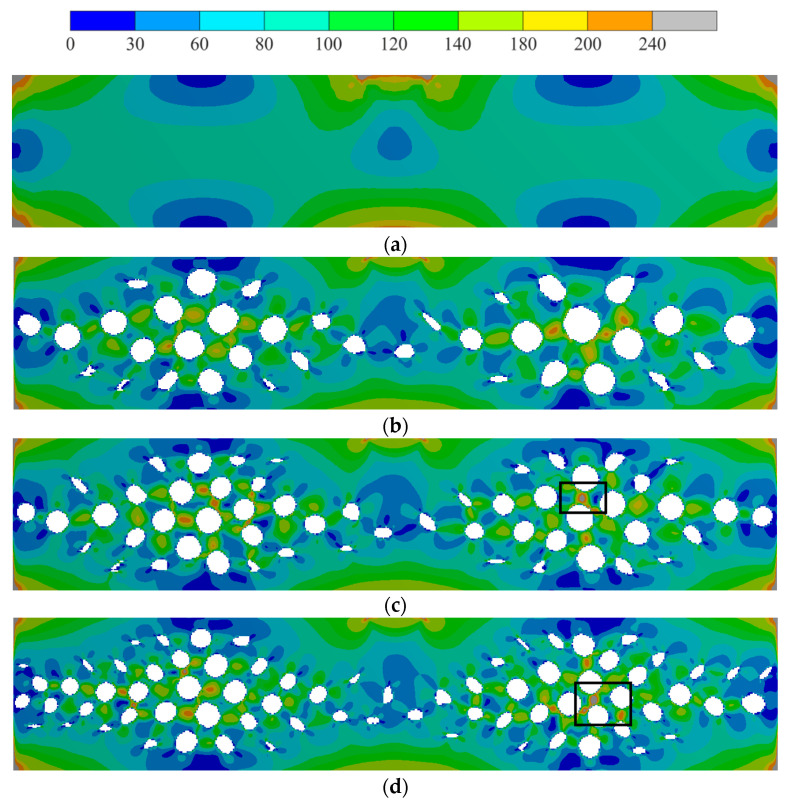
Von Mises stress field distribution, MPa: (**a**)—original structure, (**b**)—smoothed structure for 50 clusters, (**c**)—smoothed structure for 75 clusters, and (**d**)—smoothed structure for 100 clusters; the stress raisers areas are highlighted by boxes.

**Figure 6 biomimetics-09-00185-f006:**
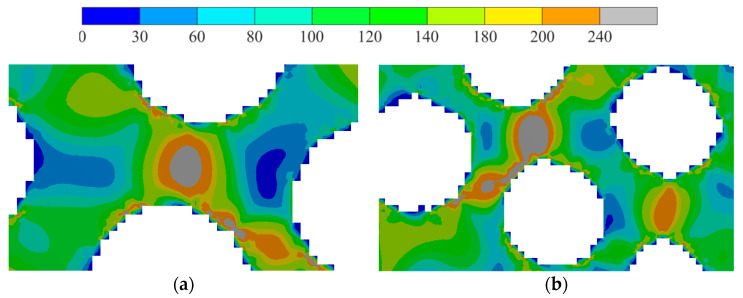
Von Mises stress field distribution, MPa: (**a**)—smoothed structure for 75 clusters and (**b**)—smoothed structure for 100 clusters.

**Figure 7 biomimetics-09-00185-f007:**
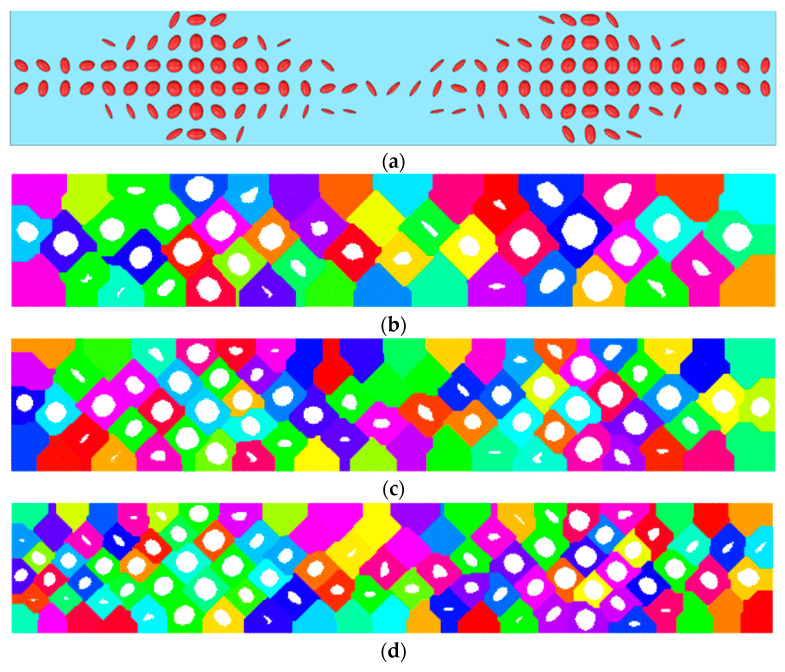
Porous structure: (**a**)—restored using the direct method, (**b**)—smoothed structure for 50 clusters, (**c**)—smoothed structure for 75 clusters, and (**d**)—smoothed structure for 100 clusters; clusters are marked with different colors.

**Table 1 biomimetics-09-00185-t001:** The values of coefficients of the approximation polynomial for stiffness parameters.

	*c* _00_	*c* _10_	*c* _01_	*c* _11_	*c* _21_	*c* _31_	*c* _12_	*c* _22_	*c* _13_
*E*_11_, GPa	109	−3.9	−5.3	−192	287	−115	319	−209	−136
*E*_22,33_, GPa	102	2.9	10.6	−111	325	−278	−17.8	−18.7	27
*G*_12,13_, GPa	10.7	−0.1	0.25	−2.7	13	−10	−3.9	−0.1	4.1
*G*_23_, GPa	2.5	−0.1	−0.06	−4.4	8	−3.4	6.4	−5	−2.5
υ_12,13_	0.011	–	–	–	–	–	–	–	–
υ_23_	0.017	–	–	–	–	–	–	–	–

where *E*_11_, *E*_22_, and *E*_33_ correspond to *C*_11_, *C*_22_, and *C*_33_, respectively; *G*_12_, *G*_13_, and *G*_23_ correspond to *C*_44_, *C*_55_, and *C*_66_, respectively; υ_12_, υ_13_, and υ_23_ correspond to *C*_12_, *C*_13_, and *C*_23_, respectively.

**Table 2 biomimetics-09-00185-t002:** Porosity values in zones for different clusterization cases.

Number of Clusters	Zones		
	I	II	III
50	5%	15%	2.8%
75	4.5%	14.8%	2.5%
100	4.3%	14.8%	2.4%

## Data Availability

The data that support the findings of this study are available from the corresponding author upon reasonable request.

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
