# Peer review of "Extension of the Voronoi Diagram Algorithm to Orthotropic Space for Material Structural Design"

_biomimetics, 2024, doi:10.3390/biomimetics9030185_

Round 1
Reviewer 1 Report
Comments and Suggestions for Authors
Clarifications on the points raised below ought to be provided before a definitive proposal regarding publication of the paper is made:
1. What is the speciality of Voronoi Diagram Algorithm as compared to other existing algorithms?
2. Why specifically the 50, 75 and 100 clusters are choosed? Why not other values?
3. Why specifically beam with fixed end faces and center kinematic loading is choosed? Why it is not created for all types of supports and loading conditions?
4. What is the mechanism behind the 75 and 100 clusters shows local stress factor appeared in smoothed structure when compared to 50 cluster?
5. Table 1 is filled only with notations. Make the table in a way to understand the readers.
6. What is the necessity of incorporating the Smoothing structure algorithm and ColorStep algorithm in your study?
7. What does the “Zone” represent in Table 2.
8. What does the range represent in Figure 5 and 6.
9. The manuscript does not go into considerable detail about how to apply and discuss the findings.
10. The conclusions are similarly lacking in information.
Author Response
I am very pleased with your comments. Thanks to you, we have an opportunity to improve the quality of the article. I hope that the changes we have made will allow us to present the study more clearly. For convenience, all changes are highlighted in color.
1.What is the speciality of Voronoi Diagram Algorithm as compared to other existing algorithms?
Once the distribution of the parameter vector is found, the next step is to reconstruct the obtained geometry. In practice, a finite element mesh is associated with a distribution of basic cells. This means that each finite element is treated as a basic cell. This is generally not true. Additionally, the problem of smoothing the microarchitecture on the basic cells borders occurs. From this point of view, the shape of each basic cell is unknown. One of the solutions is to use parameter vector's distribution to solve this problem. The study used clustering method for porosity distribution. Next, pore shape should be restored in each cluster according to distribution of ellipticity values and directions. For these purpose the Voronoi diagram for orthotropic space was generalized. The corresponding paragraph has been added to the article.
2. Why specifically the 50, 75 and 100 clusters are choosed? Why not other values?
During the numerical simulations, the authors tried different numbers of clusters. The value from 50 to 100 was suitable. Of course, this raises relevant questions of how to choose the number of clusters for smoothing. More research is needed to find the optimal number or appropriate criteria. The corresponding text was added.
3. Why specifically beam with fixed end faces and center kinematic loading is choosed? Why it is not created for all types of supports and loading conditions?
This is just a model task. The proposed algorithm can be used to any type of boundary conditions. The specific beam with fixed end faces and center kinematic loading was chosen because for this task parameter vector results are known and published. So, the general pipeline can be formulated to restore geometry. The first step is to formulate the boundary value problem for the stress-strain state. Then it is necessary to find the parameter vector for this problem (the method given in the article [32] or another one can be used). The last step is to apply the proposed Voronoi diagram algorithm to the parameter vector distribution. The corresponding paragraph has been added to the article.
4. What is the mechanism behind the 75 and 100 clusters shows local stress factor appeared in smoothed structure when compared to 50 cluster?
The reconstructed smoothed geometry was simulated in the same problem formulation (15)-(24). And local stress factor were found using solving strain-stress problem for smoothed geometry. The corresponding text was added.
5. Table 1 is filled only with notations. Make the table in a way to understand the readers.
Table description was expanded.
6. What is the necessity of incorporating the Smoothing structure algorithm and ColorStep algorithm in your study?
The Smoothing structure algorithm is an algorithm for extending the Voronoi diagram to an orthotropic space. And ColorStep algorithm is a used in Smoothing structure algorithm. Both are the main results of the study.
7. What does the “Zone” represent in Table 2.
The zones are shown in Figure 4 (zones marked by roman numerals I, II, III) and porosity values in the zones are given in Table 2. The corresponding text was added.
8. What does the range represent in Figure 5 and 6.
Von Mises stress field distribution (values in MPa) is shown in Figures 5 and 6. The corresponding text was added.
9. The manuscript does not go into considerable detail about how to apply and discuss the findings.
The discussion and conclusion have been extended to give more emphasis on the application of the findings.
10. The conclusions are similarly lacking in information.
The conclusion was expanded.

Reviewer 2 Report
Comments and Suggestions for Authors
This manuscript, the author described the Voronoi diagram algorithm to demonstrate the center kinematic motions. Although the author mentioned some simulation result, I cannot find the current manuscript is suitable to be accepted in this journal. My comments are below:
1, The author described the porosity which was increased from 21.5% to 22.8%, I think it is too small, what's the main idea of this research??
2, Please describe more of boundary condition in (15)-(20).
3, In Fig. 4, it looks similar between 75 and 100 clusters, what's the difference of this figure?
4, Please re-write the conclusion section. There are only mention the main result.
Author Response
I am very pleased with your comments. Thanks to you, we have an opportunity to improve the quality of the article. I hope that the changes we have made will allow us to present the study more clearly. For convenience, all changes are highlighted in color.
1. The author described the porosity which was increased from 21.5% to 22.8%, I think it is too small, what's the main idea of this research??
I am afraid that the essence of the study was misunderstood. If we are talking about porosity, the initial (solid) region should be compared with the smoothed area. This means that porosity increases from 0 to 21.5 – 22.8 %. And the study focuses on how to recover the geometry according to the distribution of structural parameters. In practice, a finite element mesh is associated with a distribution of basic cells. This means that each finite element is treated as a basic cell. This is generally not true. Additionally, the problem of smoothing the microarchitecture on the basic cells borders occurs. From this point of view, the shape of each basic cell is unknown. One of the solutions is to use parameter vector's distribution to solve this problem. The study used clustering method for porosity distribution. Next, pore shape should be restored in each cluster according to distribution of ellipticity values and directions. Thus, the main idea of this study is to generalize the Voronoi diagram for orthotropic space. The corresponding text was added.
2. Please describe more of boundary condition in (15)-(20).
The description of boundary condition was extended – equations (21)-(24).
3, In Fig. 4, it looks similar between 75 and 100 clusters, what's the difference of this figure?
The difference is in the pore distribution. The displacement fields are almost the same in all cases (a-d in Fig. 4). This is as it should be. The difference arises only in the von Mises stress field distribution (in Figure 5). The corresponding text was added.
4, Please re-write the conclusion section. There are only mention the main result.
The conclusion was expanded.

Reviewer 3 Report
Comments and Suggestions for Authors
The manuscript presents an adaptation of the Voronoi diagram algorithm for use in orthotropic spaces, aimed at designing the structure of materials. The method requires inputs such as porosity, ellipticity, and the direction of ellipticity.
The manuscript is challenging to follow and comprehend. To improve its quality, the reviewer suggests addressing the following points:
· Proofread the article for any errors.
· Provide a clearer explanation of the differences between the proposed adaptation and the original method.
· Specify the platform on which the algorithm was implemented and tested.
· The model task needs a more detailed description, including where it was implemented and the analyses conducted.
· Section 3.2 requires clearer explanations, as it is currently difficult to follow and understand.
Implementing these suggestions would give readers a more comprehensive understanding of the methods described and their potential applications.
Comments on the Quality of English Language
The manuscript is challenging to follow and comprehend
Author Response
I am very pleased with your comments. Thanks to you, we have an opportunity to improve the quality of the article. I hope that the changes we have made will allow us to present the study more clearly. For convenience, all changes are highlighted in color.
· Proofread the article for any errors.
The text was revised.
· Provide a clearer explanation of the differences between the proposed adaptation and the original method.
The text has been expanded (especially Materials and Methods, Discussion and Conclusion sections)
· Specify the platform on which the algorithm was implemented and tested.
The smoothing structure algorithm was implemented in the Matlab software. Stress-strain problem was solved in Ansys software. The corresponding text was added.
· The model task needs a more detailed description, including where it was implemented and the analyses conducted.
The model problem was chosen because the authors previously had results for the same problem. The developed algorithm can be easily applied to any stress-strain problem. The text has been revised and supplemented for better understanding.
· Section 3.2 requires clearer explanations, as it is currently difficult to follow and understand.
The text of section 3.2 has been revised. In addition, the Discussion has been expanded.

Reviewer 4 Report
Comments and Suggestions for Authors
This paper studied the influences of arrangements of porosity in structures on mechanical properties and equivalent stresses of structure by employing mathematical formulation. Based on reviewer knowledge, the paper is interesting and has practical applications for various industries. I strongly recommend it for publication after minor revision as the following:
1) Can we consider any porous structure like auxetic structure as opened or closed cell foam? please explain completely.
2) The validation study is needed. How can authors trust to their model and results? Please explain.
3) Why didn’t authors fabricate their model via 3D printer and compare the results of theoretical formulation and experimental tests?
4) There are some typo and grammatical errors in the manuscript. Please polish it carefully.
5) The paper entitled "Functionally graded saturated porous structures: A review " is suitable for your work. Please discuss about it in the introduction.
Comments on the Quality of English Language
Minor editing of English language required
Author Response
I am very pleased with your comments. Thanks to you, we have an opportunity to improve the quality of the article. I hope that the changes we have made will allow us to present the study more clearly. For convenience, all changes are highlighted in color.
1) Can we consider any porous structure like auxetic structure as opened or closed cell foam? please explain completely.
The algorithm based on the Voronoi diagram concept was implemented for designing an irregular structure. Any structure can be chosen as a model problem. But it should be described by parameters of porosity, ellipticity and its direction.
2) The validation study is needed. How can authors trust to their model and results? Please explain.
The main result of this study is the generalization of the Voronoi diagram to orthotropic space. The validation of the study is illustrated in a test problem. The presented model problem illustrates the practical use of the proposed algorithm. Of course, there remain problematic issues such as the choice of the number of clusters, stress raiser, etc. And that's the motivation for future research.
3) Why didn’t authors fabricate their model via 3D printer and compare the results of theoretical formulation and experimental tests?
As it was mentioned earlier, the main result of this study is the generalization of the Voronoi diagram to orthotropic space. The presented model problem illustrates the practical use of the proposed algorithm. And fabrication of models via 3D printer will be presented in future research.
4) There are some typo and grammatical errors in the manuscript. Please polish it carefully.
The text was revised.
5) The paper entitled "Functionally graded saturated porous structures: A review " is suitable for your work. Please discuss about it in the introduction.
The paper was added.

Round 2
Reviewer 2 Report
Comments and Suggestions for Authors
The author revised significantly and I suggest it would be published in this Journal.
Reviewer 3 Report
Comments and Suggestions for Authors
The author made all the requested corrections, therefore the article can be accepted